

# Aerosol-landscape-cloud interaction: Signatures of topography effect on cloud droplet formation

Sami Romakkaniemi[1], Zubair Maalick[2], Antti Hellsten[3], Antti Ruuskanen[1], Olli Väisänen[2], Irshad Ahmad[2], Juha Tonttila[1,4], Santtu Mikkonen[2], Mika Komppula[1], and Thomas Kühn[1,2]

[1]Finnish Meteorological Institute, Kuopio, Finland

[2]Department of Applied Physics, University of Eastern Finland, Kuopio, Finland

[3]Finnish Meteorological Institute, Helsinki, Finland

[4]Karlsruhe Institute of Technology, Karlsruhe, Germany

Correspondence to:

Sami Romakkaniemi

(sami.romakkaniemi@fmi.fi)



**Abstract**

Long-term in situ measurements of aerosol-cloud interactions are usually performed in measurement stations residing on hills, mountains, or high towers. In such conditions, the surface topography of the surrounding area can affect the measured cloud droplet distributions by increasing turbulence or causing orographic flows and thus the observations might not be representative for a larger scale. The objective of this work is to analyse, how the local topography affects the observations at Puijo measurement station, which is located in the 75 m high Puijo tower, which itself stands on a 150 m high hill. The analysis of the measurement data shows that the observed cloud droplet number concentration mainly depends on the CCN concentration. However, when the wind direction aligns with the direction of the steepest slope of the hill, a clear topography effect is observed. This finding was further analysed by simulating 3D flow fields around the station and by performing trajectory ensemble modelling of aerosol- and wind-dependent cloud droplet formation. The results showed that in typical conditions, with geostrophic winds of about 10 ms$^{-1}$, the hill can cause updrafts of up to 1 ms$^{-1}$ in the air parcels arriving at the station. This is enough to produce in-cloud supersaturations higher than typically found at the cloud base (SS of ~0.2%), and thus additional cloud droplets may form inside the cloud. In the observations, this is seen in the form of a bi-modal cloud droplet size distribution. The effect is strongest with high winds across the steepest slope of the hill and with low liquid water contents, and its relative importance quickly decreases as these conditions are relaxed. We therefore conclude that, after careful screening for wind speed and liquid water content, the observations at Puijo measurement station can be considered representative for clouds in a boreal environment.

## 1. Introduction

Atmospheric aerosol particles that, at a given supersaturation, are big and hygroscopic enough to form cloud droplets, are termed cloud condensation nuclei (CCN). Spatial and temporal changes in the amount of CCN particles affect the number concentration of cloud droplets (CDNC) and with it the cloud properties, which in turn may lead to changes in the atmospheric radiative transfer and hydrological cycle. Despite decades of research efforts, the interactions between aerosols and clouds are still considered to make one of the biggest contributions to the uncertainties in the estimates of anthropogenic radiative forcing (Boucher et al., 2013).

To validate remote observations and large scale modelling results of aerosol-cloud interactions, reliable and representative direct measurements of the aerosol effect on cloud microphysics are needed. In practice, aerosol-cloud interactions can be measured *in situ* using airplanes (e.g. Gillani et al., 1995; Twohy et al., 2001; Stevens et al., 2003; Romakkaniemi et al., 2009; Wood et al., 2011), surface based on hills and mountains (e.g. Wodbrock et al, 1994; Baltensberger et al., 1998; Komppula et al., 2005; Asmi et al., 2012), or from high towers (Portin et al., 2009). All of these methods have limitations related to spatial or temporal representativeness.





In so-called hill-cloud studies, the topography affects the measurements, as winds along the slope of the hill cause upward motion, which leads to the activation of new aerosol particles to form cloud droplets. Observations of cloud properties in such setups can therefore not be generalized to free atmospheric clouds. Instead, the results from such measurements are mainly used to study the partitioning between interstitial aerosols and cloud droplets, i.e to study how aerosol composition or

size affects cloud droplet formation in certain conditions. In some well-characterized Lagrangian measurement setups, by tracking air masses through several measurement stations, it is also possible to study how aerosol properties are modified by in-cloud chemistry (e.g. Bower et al., 1999; Hermann et al., 2005; Henning et al., 2014).

At mountain stations, for instance at Jungfraujoch, several studies indicate a clear correlation between wind speed and cloud droplet number concentration (Hammer et al., 2014; 2015). Thus the measured cloud droplet properties can be generalized to

a larger scale only after careful data screening to account for local conditions. In airborne measurements such problems can be avoided by choosing measurement areas that are representative also for larger scales, but here the shortcoming is the high cost, which makes long-term measurements difficult to fund and thus the temporal coverage is limited.

In hill cloud studies, the updraft caused by the hill is strongest near the surface and decreases as a function of altitude. Thus the effect of the local topography on observations could be decreased if the measurement station would be located on a high

tower. In such a measurement setup, it depends on the height of the tower, the height and slope of the hill, and the wind speed, whether the observation can be generalized to a larger area or not.

Here we study the dependence of measured cloud droplet number concentrations on the local topography at the Puijo measurement station in Kuopio, Finland (Leskinen et. al., 2009). The station is located in the 75 m high tower residing on Puijo hill, which itself rises 150 m above the surrounding lake level. Previously, the measurements from the tower have been

mainly used to study how the aerosol particle composition and concentration affects the cloud droplet number concentration (e.g. Portin et al., 2009; 2014; Hao et al., 2013). Now we explore if the measurement conditions can be considered representative for a boreal environment on a larger scale. To this end we analyse a four-year-long time series of continuous measurements of aerosol, cloud, and atmospheric properties. The analysis includes simulations of flow dynamics and detailed aerosol microphysics using detailed topography data of the area surrounding the tower.

**2. Methods**

**2.1 In situ measurements at Puijo station**

Puijo measurement station resides on the top floor of the Puijo observation tower (62°54'34'' N, 27°39'19'' E, 306 m above sea level and 224 m above the surrounding lake level). As can be seen in Fig. 1, Puijo tower itself is 75 m high and is located on the top of Puijo hill. The surroundings of the station contain smaller hills and large lake areas. Also, residential areas of

different sizes surround the tower, with the biggest in the east and south and smaller in the southwest, west, and northwest.



All local aerosol sources are located within 10 km from the tower at an approximately 200 m lower altitude than the measurement level. A more detailed overview of the station and the surrounding area can be found in Leskinen et al. (2009).

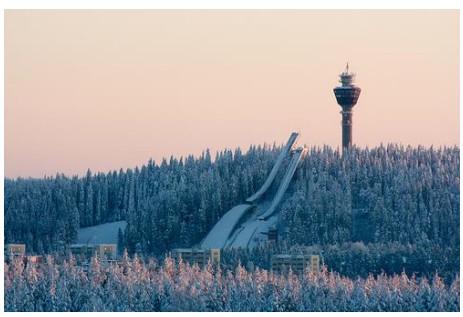
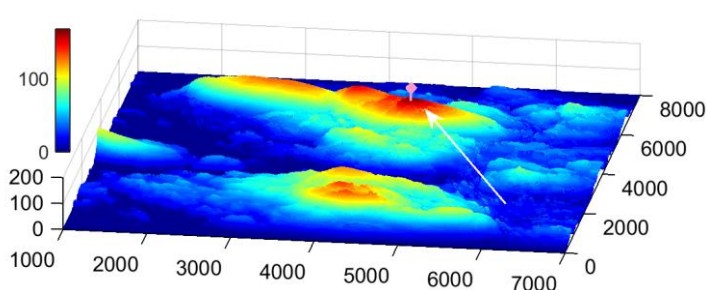

Figure 1: Photograph of tower (a) and map of topography (b)

Cloud droplets were observed with a cloud droplet probe (CDP, Droplet Measurement Technologies) with a 10 s time resolution during long term measurements and 1 s time resolution during special intensive campaigns. From the CDP we get information of the droplet size distribution between 3 and 50 μm, with 1 μm size resolution up to 14 μm, and 2 μm for larger droplet sizes. The CDP at Puijo tower is mounted on a swivel, which keeps the inlet facing the wind. It also has a custom-

built tubular inlet with an external pump to provide a constant sample flow (13 ms$^{-1}$, verified with an external anemometer), which is needed to estimate the particle concentration from the number of particle counts. The accuracy of the CDP is estimated to be 20–30 %, which is typical also for other devices with the same detection principle (e.g., forward-scattering spectrometer probe, FSSP) (Brenguier and Bourrianne, 1998). The CDP data were also used to estimate the cloud liquid water content (LWC) by calculating the total volume of the droplet population.

The particle size distribution in the size range between 3 and 800 nm is measured with a twin differential mobility particle sizer (twin-DMPS) (Winklmayr et al., 1991; Jokinen and Mäkelä, 1997). The instrument is connected to the twin inlet system (one inlet for interstitial particles and another for total aerosol including also the cloud droplet residuals) at all times and a full size distribution for both sampling lines is provided with a 12 min time resolution. Although the sampling system allows differentiation between total and interstitial particle populations, here we only use the information from the total

aerosol line, as this is the quantity that affects cloud droplet formation.

The basic weather parameters, such as temperature (Vaisala HMT337), relative humidity (Vaisala HMT337), amount of precipitation (Vaisala FD12P), and wind speed and direction (Thies UA2D) are measured continuously at the roof of the tower. All weather instruments are located approximately 2 m above the roof of the tower except for the anemometer, which is mounted on a mast at a height of 5 m above the roof in order to decrease the effect of the tower on the measured winds.

The data from these instruments is saved as one-minute averages.



The CDP data were restricted to be valid only during low-level cloud events. The cloud base height was estimated based on the calculated liquid water content and the possibility for broken clouds was additionally excluded through visibility measurements. The numerical value for the minimum LWC was set to 0.02 $\mathrm{gm^{-3}}$. Additionally, the LWC was constrained to a maximum value of 0.25 $\mathrm{gm^{-3}}$, as for larger LWC values the cloud droplets become so large, that the cloud can no longer be classified as non-precipitating.

## 2.2 Modelling

The flow fields around Puijo hill and Puijo tower were modelled using the Large Eddy Simulation (LES) model PALM (Raasch and Schröter, 2001; Maronga et al., 2015). The LES domain covered a 15 km x 8 km area around Puijo tower and extended up to 1 km height, while the boundary layer depth was about 370 m from the lake level. The wind direction was chosen to be 205° for reasons explained in the observations section. The geostrophic wind speed was set to 10.8 $\mathrm{ms^{-1}}$. Neutral stratification was set inside the entire boundary layer, and a capping inversion of 0.02 $\mathrm{Km^{-1}}$ starting from $z=340$ m was set through the initial conditions in order to prevent the boundary layer from growing in height during the simulation, since quasi stationary conditions were needed for the analysis. To resolve the turbulence caused by the surface, the LES grid spacing was 5 m in the mean wind direction and 4 m in both crosswind and vertical directions. The terrain topography needed was obtained from the National Land Survey of Finland, which has a spatial resolution of 2 m. The surface boundary condition for momentum was modelled using the logarithmic law of the wall for neutrally stratified conditions

$$u_* = \frac{\kappa v_{tan}}{\ln(\Delta p/z_0)},$$

where $u_*$ is the local instantaneous surface friction velocity, $v_{tan}$ is the local instantaneous surface-tangential velocity, and $\Delta p$ is the Prandt-layer thickness, which in this case is half of the height of the first computational level above the local ground surface. The roughness length is denoted by $z_0$ and $\kappa$ is the von Kármán constant.

Puijo hill is mostly covered by old forest containing spruce and pine trees, and their effect was taken into account by adding a constant displacement height, $z_d$, of 17 m to the local terrain height and using a constant $z_0$ of 3 m for the surface boundary conditions for forested areas. These values of $z_d$ and $z_0$ were approximated as representative for coniferous forest following Jarvis et al. (1976). The effects of urban areas were taken into account similarly using estimated values of $z_d=12$ m and $z_0=3$ m.

Approaching turbulent wind flow on the inflow boundary was modelled as follows. A precursor simulation was first performed, with a flat terrain but otherwise similar setup as for the actual simulation. The mean profiles of the prognostic variables were stored and used on the inflow boundary of the actual simulation. In the actual simulation, turbulent fluctuations as functions of vertical and cross wind directions and time were recycled from a plane 2800 m downwind from the inflow boundary, and added to the inflow mean profiles on the inflow boundary. This way a realistic turbulent inflow boundary condition was created. The actual simulation was initialized using the results of the precursor simulation.



Because the version of PALM used here does not model aerosol processes or aerosol-cloud interactions, a separate cloud parcel model was used to simulate the cloud droplet formation close to Puijo tower. The model used has been described elsewhere (Kokkola et al., 2003), and it has been used in several aerosol-cloud interaction studies (e.g., Romakkaniemi et al., 2005; 2011; 2012). In order to estimate how topography affects the air parcel history before it is measured, the model was run along trajectories obtained from the PALM simulation. We have used a similar approach earlier in Romakkaniemi et al. (2009) to study the aerosol effect on stratocumulus clouds observed over the east coast of the UK and in Romakkaniemi et al. ( 2014) to study how semi-volatile aerosol particles grow by water uptake in a CCN counter, a device to measure aerosol CCN activity. However, as the PALM simulations were conducted assuming a dry atmosphere, we here assumed the trajectories to be adiabatic. The modelled wind fields were saved with a very high temporal resolution for a selected time period. As the endpoint of the trajectories needed to be at the measurement station, the trajectories were calculated backwards in time, with a constant difference of two seconds in arrival time between trajectories. Owing to the high amount of data storage required to store the 3D wind fields, which are needed to calculate the trajectories, the length of the trajectories was limited to about 5 km.

During the initialization phase of the simulation, the parcels were lifted from the local lake surface up to the initial height of the trajectory using a constant updraft of 0.15 ms$^{-1}$. In cases where the initial position was within the cloud, this updraft produced cloud droplet concentrations that are typical for the clouds observed at Puijo tower. The aerosol size distributions needed to simulate the aerosol effect on cloud droplet formation were obtained from the DMPS-measurements at Puijo tower. We used an aerosol composition and mixing state that is typically observed with AMS (e.g. Hao et al., 2013; Portin et al., 2014) and HTDMA (e.g. Väisänen et al., 2016) at Puijo. To represent aerosol particles in the model, 600 size sections were used in order to properly resolve the activation kinetics.

## 3. Results

### 3.1 Measurement data.

For the analysis we used data collected at Puijo station between 1 October 2010 and 30 November 2014. Because we do not have continuous measurements for the aerosol composition or amount of CCN in different supersaturations, we used the number concentration of aerosol particles larger than 100 nm in diameter ($N_{100}$) as a proxy for the CCN concentration. The same size limit has been used also in previous studies and was found to be a reasonable proxy for aerosol activation in low altitude clouds (Portin et al., 2009; Asmi et al., 2012; Ahmad et al., 2013).







Figure2: CCN dependence on wind speed and direction: (a) CDNC, (b) $N_{100}$ , (c) number of CDNC samples  (d) Topography of the area around Puijo hill, with the wind direction used in the simulations marked as red arrow. (e) Wind speed trend ($\partial CDNC/\partial v$) for different wind direction intervals and (f) standardised wind speed and $N_{100}$ trends.





Figure 2a shows the average CDNC values measured at Puijo tower as function of wind speed and wind direction. The figure shows a general trend of increasing CDNC with increasing wind speed, if the wind arrives from southern directions. For winds arriving from northern directions, on the other hand, CDNC decreases with increasing wind speeds. For a fixed wind speed, the CDNC values are maximal for wind directions of about 180° to 210°. To a great extent the CDNC dependence on

wind speed and direction can be explained with the corresponding values of our CCN proxy, $N_{100}$ (Fig. 2b). Air masses arriving from the north pass over regions with very little anthropogenic emissions and $N_{100}$ values are very low to start with. The higher the wind speed, the less time the air masses spend over land and thus $N_{100}$ concentrations decrease (Väänänen et al., 2013), which directly reflects in the CDNC values. On the other hand, winds arriving from the south contain central European and continental air masses resulting in much higher $N_{100}$ values than for winds coming from the north. Thus

CDNC values are in general higher as well.

However, as the air masses arrive at Puijo hill, they experience updraft, which depends on the horizontal wind speed and the slope of the hill. As the slope of the hill varies a lot, it is difficult to estimate when this topography effect is important for cloud droplet formation and when not. We therefore performed a multivariate analysis of covariance (ANCOVA) to discriminate between the CCN and topography effects on CDNC. As predictors for CDNC we used the wind conditions

(wind speed and direction separately) and $N_{100}$ concentration. In order to avoid biases caused by inadequate fit functions (Pitkänen et al., 2016), we used a robust regression fit function (rlm; Yegorov, 2016) in the analysis. Figure 2e shows the CDNC trend (which can be interpreted as $\partial CDNC / \partial v$) due to variations in wind speed for wind direction intervals of 30°. The bars in the plot denote confidence intervals. We found the strongest effect (~ 15 $cm^{-3}/ms^{-1}$) of wind speed on CDNC for wind directions around 210°, with further significantly positive trends for winds from 120°, 150°, and 180°. For all other

directions, the wind speed trend is very close to zero or even negative. The negative trends, however, are not representative due to the sparseness of the data for these directions (see Fig. 2c). A very drastic drop in wind speed trends can be seen between 210° and 240°. This can be explained by an equally drastic change in topography and surface type. Air masses arriving at Puijo from 240° pass first over another forested hill and then over a lake. The descent that the air undergoes while moving over the hill may cause a large number of droplets to evaporate. This will be explained in detail later.

In order to be able to directly compare the relative importance of CCN and wind speed on CDNC, we repeated our analysis on standardised $N_{100}$ and wind speed distributions (Fig. 2f). The plotted quantities are physically not as easily interpretable as the trend analysis in Fig. 2e, but this representation enables us to directly compare the relative influence of the two predictors ($N_{100}$ and wind speed) on CDNC. The plot shows a positive $N_{100}$ trend everywhere, except for one point (120°). This special direction lies in the wake of the city, where freshly emitted aerosols, which are much less effective CCN, may make up a

considerable portion of the $N_{100}$ concentration. In all other points, it is quite clearly visible that CCN has a larger influence on CDNC than wind speed, although confidence is much lower for northern directions. However, for the southern directions the wind speed trend is both significant and comparable in magnitude to the $N_{100}$ trend. We therefore performed a numerical modelling study to further quantify the effect that wind speed has on measured CDNC through the local topography.





## 3.2 Large eddy modelling

For a better understanding of the topography effect on the vertical winds and cloud droplet formation, we performed PALM simulations using a wind direction of 205°, as there low-level clouds are most often observed. This is also the wind direction where the effect of wind speed on CDNC seems strongest in the measured data. At the imposed wind speed of 10.8 ms$^{-1}$, the highest updraft caused by the hill is around 2 ms$^{-1}$ close to the surface (Fig. 3). However, as the station is located 75 m above the surface, the air masses arriving at the station are not subject to the strongest updrafts (Fig. 3). To gain more insight on the path that the air parcels typically travel before being measured at the station, we extracted trajectories from the LES simulation (see Methods section for details). Altogether we extracted more than 100 trajectories, of which a selected set with diverse properties is presented in Fig. 3. Although all parcels experience uplift before arriving at the station, the path and thus the strength of the updraft varies. During the last 20 s before the air parcels arrive at the measurement station, we find that the mean updraft velocity is 0.76 ms$^{-1}$, with minimum and maximum values of 0.47 ms$^{-1}$ and 1.01 ms$^{-1}$, respectively.

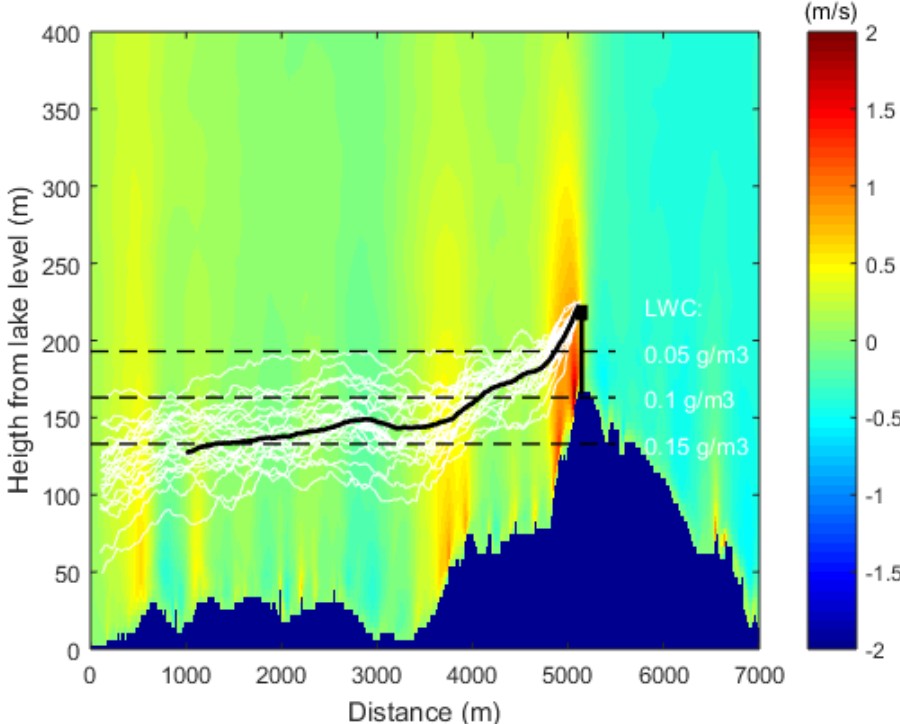

Figure 3: Simulated average wind fields and example trajectories (white curves) of air parcels. The black line depicts the average over all simulated trajectories.





### 3.3 Parcel modelling

The trajectories were used as input for an air parcel model to see how the updraft affects droplet formation. Aerosol properties needed to simulate cloud droplet formation where taken from DMPS measurements, with the aerosol composition (hygroscopicity) following the typical composition measured during different campaigns (Hao et al., 2013; Portin et al. 2014; Väisänen et al. 2016) at the station. The size distribution used here was bimodal with mean geometric diameters of 60 and 230 nm, and number concentrations of 480 and 190 cm$^{-3}$ in Aitken and accumulations modes, respectively. This kind of bimodal aerosol size distribution is typical for cloudy boundary layers observed at Puijo and also other similar measurements (Komppula et al., 2009, Portin et al., 2014). With the modelled size distribution, the value for $N_{100}$ is 320 cm$^{-3}$. For simplicity, the aerosol was assumed to be composed of ammonium sulphate and a non-soluble compound, which were mixed with mass fractions of 1/3 and 2/3, respectively. Note here that, as the motivation of these numerical simulations was to study the topography effect on CDNC, the aerosol size distribution and composition is the same in all simulations and thus the CCN effect on CDNC is not visible.

As the version of PALM used here does not model clouds, we assumed the air parcel to follow adiabatic trajectories, and different cloud base heights where used to simulate different liquid water contents (LWC) at the measurement station. The minimum LWC was 0.05gm$^{-3}$, corresponding to a measurement altitude of ~40 meters from the cloud base in adiabatic conditions. Measurements with lower LWC are usually omitted from the data analysis at Puijo station, because the probability for broken clouds, which cannot be distinguished with our DMPS due to its 6 min scan time, increases. Additionally, with high cloud droplet concentrations the mean droplet size at low total LWC is in the lowest measurement channels of our CDP, where its measurement uncertainty is highest.



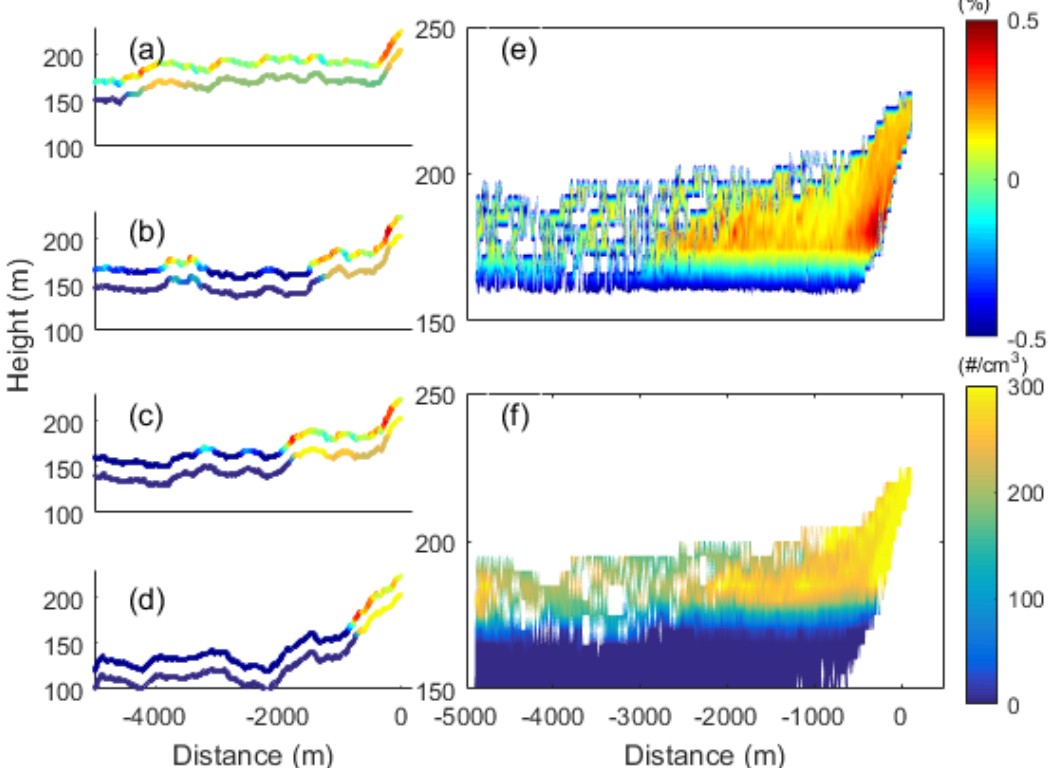

Figure 4: a-d) Selected trajectories with supersaturation and CDNC superimposed. The two curves in each plot show the same trajectory, but are offset in y-direction for easier distinction. The upper curve shows the correct height. e) Average supersaturation over all simulated trajectories. f) Average CDNC over all simulated trajectories.

The different trajectories seen in Fig. 3 have very different supersaturation histories before being measured at the station. This can be seen in Fig. 4a-d, where four highly diverse example trajectories are presented. The air parcels reach the maximum modelled supersaturation of ~0.2% approximately 10 m above the cloud base. This corresponds nicely with the observations, where the effective supersaturation has been observed to vary between 0.16% and 0.29% (Väisänen et al.,

10  2016). After the first activation of droplets at the base of the cloud, changes in the sign of the updraft are reflected as a change of sign in the supersaturation. This can also lead to changes in CDNC. If the supersaturation is negative or close to zero, the smallest droplets may evaporate (Wood et al., 2002; Romakkaniemi et al., 2008), which may thus lead to a decrease in CDNC. The opposite can happen as well: if the saturation ratio inside the cloud is higher than the critical supersaturation of the most potential interstitial CCN, new droplets may form. However, supersaturation is quickly depleted as water

15  condenses onto pre-existing droplets and hence also the time available for an aerosol particle to exceed its critical size is





important. Thus, in order to cause such secondary activation inside the cloud, the updraft inside the cloud must be much higher than the updraft observed at the cloud base (i.e. during initial activation).

As can be seen from Fig. 4a-d, both evaporation and formation of droplets take place along the different trajectories simulated. For the trajectories presented, the updraft at the cloud base is typically ~0.15-0.2ms$^{-1}$ (leading to supersaturation of ~0.2%) whereas an updraft of ~0.6-0.8 ms$^{-1}$ or higher, depending on the liquid water content, is needed inside the cloud to exceed the cloud base supersaturation and to activate new droplets. As can be seen from Fig. 3, such high updrafts are actually simulated and can cause supersaturations as high as 0.4% (Fig. 4 e) for a short period of time. In these cases, new droplets form also inside the cloud. In cases where the air parcel has experienced downdraft, and thus evaporation of the smallest cloud droplets, before reaching the updraft again, the in-cloud activation takes place more easily due to the presence of more CCN active interstitial particles. The most striking feature of Figs. 4e and f is the very pronounced high supersaturation at the foot of Puijo hill (Fig. 4e) and the resulting strong increase in CDNC (Fig. 4f) as the air parcels climb the slope to the measurement station. However, if the cloud base is 20 m lower, corresponding to an increase in LWC of about 0.025 gm$^{-3}$ LWC at Puijo station, the modelled supersaturation is lower and the enhancement in CDNC would become less obvious (not shown). The in-cloud supersaturations may also be higher than at the cloud base already before the cloud parcel reaches Puijo hill. However, as the aerosol size distribution is bimodal (due to earlier cloud processing), the in-cloud supersaturation can cause only a limited increase in CDNC before particles from the accumulation mode are depleted. In order for the Aitken mode particles to activate, much higher supersaturations are needed. Beyond their smaller size, the particles in the Aitken mode usually also have a lower hygroscopicity and thus their CCN potential is low (Väisänen et al., 2016).

The discussed secondary activation of cloud droplets leads to a bimodal cloud droplet size distribution. Figure 5 compares the simulated, average droplet size distribution to the averaged measured size distribution. From Fig. 5b, which only averages over trajectories with bimodal size distribution (n=40), we can see that the small, in-cloud activated droplets can grow to sizes around 5 to 7 μm, whereas the larger droplets reach sizes from 8 to 12 μm. If the average is performed over all simulated trajectories (n=140), the second (left) peak decreases in height and smears out, while the first (right) peak increases in height, but the position of both peaks stays discernible (Fig. 5c). The same can also be seen in the average over the measured cloud droplet size distributions (n=3500) when air masses with wind speeds around 10 ms$^{-1}$ arrive from the same direction we used in our simulations (Fig. 5a). Note here that, due to the computational burden of the PALM simulations, the simulated trajectories are restricted to a small time window and that increasing the amount of simulated trajectories may very well increase the ensemble diversity. It is therefore not surprising that the measured and simulated curves agree qualitatively (position of peaks), but not quantitatively.





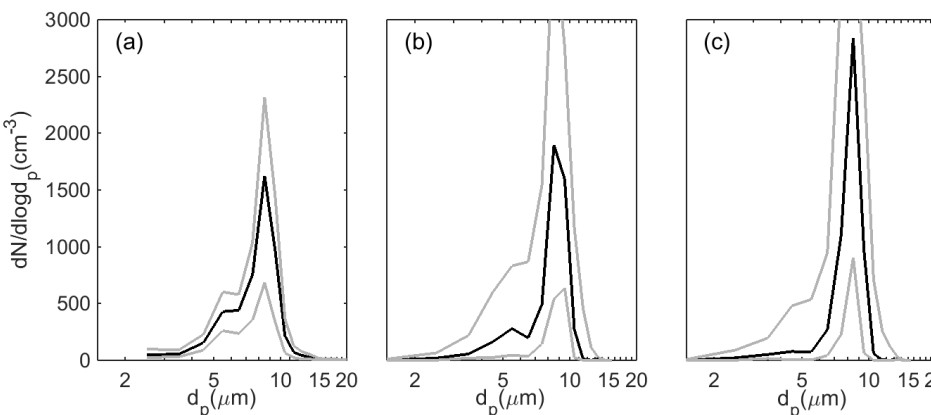

Figure 5: a) Measured cloud droplet distribution. b) Simulated cloud droplet size distributions using only trajectories that lead to second activation. c) Simulated cloud droplet size distribution using all trajectories. Black lines denote the ensemble median, while grey lines denote the 10th and 90th percentile.

To analyse why the observed enhancement in CDNC is strongest for low liquid water contents, i.e. when we are measuring close to the cloud base, we performed simulations also with different amounts of total water content by changing the initial relative humidity of the air parcels. The results support the observations. With higher amounts of liquid water, the updraft needed to create high supersaturations and secondary activation increases, because larger pre-existing droplets deplete the supersaturation more efficiently. However, if there are big enough interstitial particles available, the secondary activation can be observed with high LWC as well. This might, for instance, happen in the cases where the air parcel originally entered the cloud with a low updraft, or where some droplets in the air parcel were evaporated in the past.

## 4. Conclusions

The effect of the local topography on the observed aerosol-cloud interaction was analysed from in situ measurements conducted at the Puijo measurement station. Changes in cloud droplet concentration with changing wind speed and direction can be seen in the measurement data. However, this is mainly caused by changes in aerosol properties originating from different source areas. Based on the statistical model used, the actual effect of wind speed on the measured cloud droplet number concentration was important only when the wind direction was along the steepest edge of the hill. Otherwise, the variability in CDNC was mainly explained by the number concentration of aerosol particles larger than 100 nm in diameter, which was used as a proxy for efficient cloud condensation nuclei concentration. The topography effect was further analysed using the large eddy simulation model PALM and an air parcel trajectory model. The simulations confirmed the observations: the updraft caused by the hill can be strong enough to increase the number of observed cloud droplets. Also, in



some special cases the in-cloud activation of aerosol particles to form new cloud droplets was observed in the vicinity of Puijo measurement station both in observations and modelling results.

In the long-term time series, the in-cloud activation of new droplets at the edge of the hill slightly affects the measured aerosol-cloud interaction at the Puijo station. These conditions might explain the high droplet number concentration occasionally measured, which are not observed in satellite data when larger areas are averaged during retrievals (Ahmad et al., 2013). At any given wind speed, the increase in CDNC due to the in-cloud activation of aerosols is stronger for low liquid water contents (LWC) and its relative importance quickly decreases as the LWC grows. Thus careful screening of cases with low LWCs and strong southern winds is needed when measured droplet concentrations are reported and used, for instance, to evaluate the performance of large-scale atmospheric models. After screening, the measurements can be considered to be representative for a larger scale, and not specific to the measurement location only.

Although the local topography affects the data analysis at Puijo station only slightly, it affects the aerosol-cloud interactions in the boundary layer. To take this effect into account in modelling efforts is complicated, because computational resources are limited and thus simplified representations of cloud droplet size distribution are usually used in atmospheric models.

## Acknowledgements

This work was supported by the Academy of Finland through Centre of Excellence program (272041) CityClim project (277664), ICINA project (285068) and S.R. academy fellowship (283031). In addition, A. R. acknowledges the financial support from Maj and Tor Nessling Foundation.

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
