# Peer review of "Aerosol-landscape-cloud interaction: Signatures of topography effect on cloud droplet formation"

_Atmospheric Chemistry and Physics, 2016_

## Referee Comment (RC1) · Anonymous Referee #1 · 22 Dec 2016

General comments:

This Paper investigates the suitability/applicability of the Puijo observation station site data for investigation of cloud-aerosol interactions that occur in the free (i.e. non orographically perturbed) atmosphere. It does this by investigating the wind direction (and hence terrain and aerosol source) dependency of cloud and aerosol properties observed over a number of years at the observing station (on a 75m tower atop a moderately sized (150m high) forested hill) and by comparing these with model sensitivity studies of the same dependencies. The measurement data set was gathered from a standard set of cloud and aerosol instrumentation, while various pre-published models were used for the latter comparison (and so having already been reviewed in various

previous publications, details and use of these models will not be commented on in this review).

Unsurprisingly, the main outcome of the analysis of the observational data set is that the cloud properties observed at this site are mainly influenced by the properties of the aerosols that act as Cloud Condensation Nuclei (CCN) within this system, and also, but to a lesser extent, by the effects of the enhanced updraughts (and hence enhanced supersaturations) experienced by the aerosol-cloud system as the airmass they are contained within approaches and rises over the Puijo hill. The effects of the complexity of the terrain profile and aerosol sources in the different wind directions were simplified by confining the model sensitivity studies to investigations for the wind direction generating the biggest orographic effect in the observations, and by using a standardised aerosol input for that direction. This also allowed identification of the conditions when the orographic effect is a maximum, and when this has to be taken into consideration when applying results from the Puijo site to non orographically influenced clouds.

Although the conclusions of this work are not particularly unexpected, I find the treatment of the observational data generally acceptable, and the initialisation and use of the LES model (PALM ) to generate trajectories along which a separate cloud parcel model was then run to investigate various sensitivities, wholly satisfactory. Previous studies in the areas of work undertaken here (including the use of the same models as used here) are well referenced, and the description of methods used in this study is clear and understandable. The conclusions are also clearly stated and correct based on the results presented. I therefore consider the outcome of this work to be useful and the paper to be worthy of publication subject to dealing with a few very minor changes/typographical corrections/clarifications (listed below).

Specific issues / comments / suggestions:

Page 3, line 13: "In hill cloud studies, the updraft caused by the hill is strongest near the

surface and decreases as a function of altitude" Suggest ".... decreases as a function of height above terrain surface" – i.e. it is localised to the hill rather than it being the actual altitude of the hill/surrounding terrain height

Page 3, line 14: change "would be located" to "were located"

Page 3, paragraph 3: just because the hill can generate additional orographic enhancements in cloud properties (generally an increased droplet number for a simple hill profile where the lifting generates increased updraughts and supersaturations) it does not mean the results are not applicable to understanding processes where similar updraughts are present in other cloud types. The issue becomes more difficult when the terrain is more complex (multiple hills/valleys, varying terrain coverage) which introduces a significant complexity and uncertainty in the supersaturation history of the ground based cloud parcels, a situation which may not be observed in the free troposphere.

Page 4 line 1: suggest change "All local aerosol sources are located within 10 km from the tower at an approximately 200 m lower altitude...." to "All local aerosol sources are located within 10 km of the tower at an altitude approximately 200 m lower ....."

Page 4, line 9: "The CDP at Puijo tower is mounted on a swivel, which keeps the inlet facing the wind". Does this swivel tilt as well as rotate? What is the average wind angle (in the vertical)? Since a tubular inlet has been fitted to the instrument to fix the sample flow (like an older FSSP) it will be crucial that the probe is rotated into the wind both in the horizontal and vertical when measuring, particularly if there is a significant vertical wind angle at the measurement site (this will be less of an issue on the tower top 75m above the hill surface). This could affect the droplet size distribution measurements. Please clarify this situation.

Page 4, line 10: The inclusion of a tubular inlet may also introduce sampling artefacts through droplet breakup on the inlet edge. This should be discussed as to why this is/is not an issue.

Page 4, lines 11-12: "The accuracy of the CDP is estimated to be 20–30 %". This needs additional clarification i.e. is this a 20-30% accuracy in sizing, counting, or what? This is particularly important in order to understand how this progresses through to the integrated cloud liquid water contents which are then calculated. State the uncertainty in the calculated LWC.

Page 4, lines 13-24: "All weather instruments are located approximately 2 m above the roof of the tower except for the anemometer, which is mounted on a mast at a height of 5 m above the roof in order to decrease the effect of the tower on the measured winds". This is an issue if the CDP is one of these" weather instruments" mounted at 2m. If the anemometer is mounted 3m above the other instrument to reduce the effects introduced by the tower itself on the wind, it follows that the other instruments are sitting in air perturbed by the tower. The mounting height of the CDP needs to be stated. Anemometer, CDP and particle sampling inlets should have been mounted at a similar height on the tower (and as close to each other as possible).

Page 4, line 25: "The data from these instruments is saved as one-minute averages." Change "is" to "are" i.e. "The data from these instruments are saved as one-minute averages."

Page 5, line 1: "The CDP data were restricted to be valid only during low-level cloud events". So is this for situations where the cloud base of low level stratus clouds advecting over the region was sufficiently low to envelop the measurement tower in cloud?

Page 5, lines 1-2 ": suggest changing: ".... and the possibility for broken clouds was additionally excluded through visibility measurements." to ".... and the potential inclusion of periods of broken clouds reduced through use of visibility measurements." Page 5, line 3: insert "cloud" into: "The numerical value for the minimum cloud LWC was set to 0.02 gm-3.

Page 5, lines 4-5: ".... for larger LWC values the cloud droplets become so large, that the cloud can no longer be classified as non-precipitating". This will be true unless

there are cases where the droplet concentration is higher (in polluted events) and the LWC is spread out over that higher droplet number. So suggest adding a line like " ... for typical droplet number concentrations observed at the measurement site in such cases" or similar.

Page 5, line 9: "...while the boundary layer depth was about 370 m from the lake level". Comment on why the BL depth was set to this value (was it validated by any measurements at the time or previously?). Also change ".... from the lake level" to "..from above the lake level"

Page 5, line 16: "using the logarithmic law of the wall for neutrally stratified conditions". Is this a correct statement (... because I am not familiar with the "wall" here) – apologies if correct! (in which case describe this a bit more)

Page 6, line 5: delete "have" in "We have used a similar approach earlier 5 in Romakkaniemi et al...."

Page 8, line 11: insert "an" in "However, as the air masses arrive at Puijo hill, they experience an updraft, which depends..."

Page 8, line 32: suggest insert "at times" into: ... both significant and at times comparable in magnitude to the N100 trend".

P10, line 3: ".....typical composition measured during different campaigns (Hao et al., 2013; Portin et al., 2014; Väisänen et al., 2016) at the station." Without reading these references in detail, was a direction dependent aerosol composition used as input here (since it was mentioned earlier that some directions included sources of aerosol. i.e. was the aerosol composition input appropriate for the wind direction chosen that maximises the orographic effects in these sensitivity tests?

As an aside, since the primary control of cloud properties comes from the properties of the aerosol available as CCN, was this whole sensitivity test repeated for a quite different aerosol input? In the extreme case the shape of the aerosol distribution input

to the model could change the activation pattern of CCN and result in a different cloud response in the sensitivity tests. However I would not expect the results to change significantly. Comments?

Page 11, lines 10-11: " changes in the sign of the updraft are reflected as a change of sign in the supersaturation." Maybe emphasise by saying "changes in the sign of the updraft (i.e. changes from updraughts to downdraughts) are reflected in a change in the sign of the supersaturation (i.e. from supersaturation to subsaturation)"

Page 12, line 8: delete superfluous "also" in ".... droplets form also inside the cloud"

Page 12, line 14: delete superfluous "already" in "The in-cloud supersaturations may also be higher than at the cloud base already before the cloud parcel reaches Puijo hill"

---

## Referee Comment (RC2) · Anonymous Referee #2 · 31 Jan 2017

This manuscript presents a well-performed numerical experiment to investigate topography effects on cloud droplet activation. The paper is, in general, well-written and properly structured, with no apparent scientific errors. My only major concern is related to potential implications resulting from this study. The authors only bring up the importance of considering topography effects when analyzing cloud-related measurement data at their measurement site (see abstract and conclusions). I think this work might have broader scientific implications, i.e. cloud activation studies in other hill-cloud sites or cloud droplet activation in general in environments having a complicated topography. The authors should shortly discuss this issue, for example at the end of their conclusions. My other, mainly minor concerns are detailed below.

[Figure]

Page 3, lines 2-5: The statement made in these lines (. . .are mainly used to study. . .) needs a reference/references.

Page 5, lines 3-5. What was the basis of setting 0.02 g/m3 as the minimum LWC level? How was the maximum LWC level of 0.25 g/m3 found? Is it general knowledge or was it obtained from some kind of sensitivity tests specific for this orography case?

Page 8, lines 2 and 17; caption of figure 2: "trend" is generally considered as a property of a time series and should not be used in describing other types of gradients. Please modify.

Page 8, line 26: Please explain what is meant by standardized quantities here.

Page 13. line 6: The reader may be a bit confused about what enhancement in CDNC means here without going back in the text. Please add a few words here to make the text more readable.

Figure 2: Panels a-d differ so much from panels e-f in this figure that I would recommend splitting Figure 2 into two separate figures (Figs. 2 and 3). Furthermore, the y-axises of panels e and f do not have a unit.

[Figure]

---

## Author Comment (AC1) · 20 Apr 2017

**Authors' response to Anonymous Reviewers**

We would like to thank both Anonymous Reviewers for their valuable comments. All the comments have been taken into account and the manuscript will be revised accordingly. Below, the reviewer's comments are written in bold and followed by authors' responses.

**Anonymous Referee #1**

**General comments:**

**This Paper investigates the suitability/applicability of the Puijo observation station site data for investigation of cloud-aerosol interactions that occur in the free (i.e. non orographically perturbed) atmosphere. It does this by investigating the wind direction (and hence terrain and aerosol source) dependency of cloud and aerosol properties observed over a number of years at the observing station (on a 75m tower atop a moderately sized (150m high) forested hill) and by comparing these with model sensitivity studies of the same dependencies. The measurement data set was gathered from a standard set of cloud and aerosol instrumentation, while various pre-published models were used for the latter comparison (and so having already been reviewed in various previous publications, details and use of these models will not be commented on in this review).**

**Unsurprisingly, the main outcome of the analysis of the observational data set is that the cloud properties observed at this site are mainly influenced by the properties of the aerosols that act as Cloud Condensation Nuclei (CCN) within this system, and also, but to a lesser extent, by the effects of the enhanced updraughts (and hence enhanced supersaturations) experienced by the aerosol-cloud system as the airmass they are contained within approaches and rises over the Puijo hill. The effects of the complexity of the terrain profile and aerosol sources in the different wind directions were simplified by confining the model sensitivity studies to investigations for the wind direction generating the biggest orographic effect in the observations, and by using a standardised aerosol input for that direction. This also allowed identification of the conditions when the orographic effect is a maximum, and when this has to be taken into consideration when applying results from the Puijo site to non orographically influenced clouds.**

**Although the conclusions of this work are not particularly unexpected, I find the treatment of the observational data generally acceptable, and the initialisation and use of the LES model (PALM ) to generate trajectories along which a separate cloud parcel model was then run to investigate various sensitivities, wholly satisfactory. Previous studies in the areas of work undertaken here (including the use of the same models as used here) are well referenced, and the description of methods used in this study is clear and understandable. The conclusions are also clearly stated and correct based on the results presented. I therefore consider the outcome of this work to be useful and the paper to be worthy of publication subject to dealing with a few very minor changes/typographical corrections/clarifications (listed below).**

**Specific issues / comments / suggestions:**

**Page 3, line 13: "In hill cloud studies, the updraft caused by the hill is strongest near the surface and decreases as a function of altitude" Suggest "…. decreases as a function of height above terrain surface" – i.e. it is localised to the hill rather than it being the actual altitude of the hill/surrounding terrain height**

We agree and changed the wording as suggested.

**Page 3, line 14: change "would be located" to "were located"**

We agree and changed the wording accordingly.

**Page 3, paragraph 3: just because the hill can generate additional orographic enhancements in cloud properties (generally an increased droplet number for a simple hill profile where the lifting generates increased updraughts and supersaturations) it does not mean the results are not applicable to understanding processes where similar updraughts are present in other cloud types. The issue becomes more difficult when the terrain is more complex (multiple hills/valleys, varying terrain coverage) which introduces a significant complexity and uncertainty in the supersaturation history of the ground based cloud parcels, a situation which may not be observed in the free troposphere.**

Yes, this is true, a similar activation for example at the cloud base can be observed and used for example in the case of cumulus clouds. However, if the in situ measurements are compared to satellite observations, the mean retrieved properties of clouds can be highly different from the observed ones, as there is a probability function for different updraft velocities and cloud processing is also affecting the measurements.

**Page 4 line 1: suggest change "All local aerosol sources are located within 10 km from the tower at an approximately 200 m lower altitude…." to "All local aerosol sources are located within 10 km of the tower at an altitude approximately 200 m lower ….."**

We agree and changed the wording as suggested.

**Page 4, line 9: "The CDP at Puijo tower is mounted on a swivel, which keeps the inlet facing the wind". Does this swivel tilt as well as rotate? What is the average wind angle (in the vertical)? Since a tubular inlet has been fitted to the instrument to fix the sample flow (like an older FSSP) it will be crucial that the probe is rotated into the wind both in the horizontal and vertical when measuring, particularly if there is a significant vertical wind angle at the measurement site (this will be less of an issue on the tower top 75m above the hill surface). This could affect the droplet size distribution measurements. Please clarify this situation.**

The swivel does not tilt. The CDP is located on a roof on a mounting of only 50 cm height. Thus the vertical wind component is quite limited at the measurement location. We added some explanation to the article.

**Page 4, line 10: The inclusion of a tubular inlet may also introduce sampling artefacts through droplet breakup on the inlet edge. This should be discussed as to why this is/is not an issue.**

This is true and this was originally thought to be the reason for the observed cloud droplet mode in smaller size than main mode. However, due to the external pump, the flow inside the inlet can be held constant at 13 m/s (which is usually greater than the wind speed). The measurement volume is small and is taken from the center of the inlet, where, under assumption of a laminar flow, no broken-up droplets from the inlet edge should reach.

**Page 4, lines 11-12: "The accuracy of the CDP is estimated to be 20–30 %". This needs additional clarification i.e. is this a 20-30% accuracy in sizing, counting, or what? This is particularly important in order to understand how this progresses through to the integrated cloud liquid water contents which are then calculated. State the uncertainty in the calculated LWC.**

Here the accuracy concerns the droplet count, the droplet size detection can be calibrated quite accurately. We changed the text accordingly. Considering the particle sizing to be precise, we can estimate the uncertainty in LWC to be about 30% as well. On the other hand, allowing for a constant error of, say, 0.5 µm in droplet sizing, then the error in LWC varies with droplet size between 130% for the smalles and 3% for the largest particles. However, as most of the liquid water is in the largest droplets, the actual error in LWC due to sizing uncertainty is still relatively small, within 20% by assuming average size of 10 µm for the droplets.

We added some discussion to the results sections on how the uncertainty in the LWC affects the analysis. In principle, the activation of interstitial particles is easier in lower LWC and vice versa, but the uncertainty in LWC will not affect the qualitative findings of the paper.

**Page 4, lines 13-24: "All weather instruments are located approximately 2 m above the roof of the tower except for the anemometer, which is mounted on a mast at a height of 5 m above the roof in order to decrease the effect of the tower on the measured winds". This is an issue if the CDP is one of these" weather instruments" mounted at 2m. If the anemometer is mounted 3m above the other instrument to reduce the effects introduced by the tower itself on the wind, it follows that the other instruments are sitting in air perturbed by the tower. The mounting height of the CDP needs to be stated. Anemometer, CDP and particle sampling inlets should have been mounted at a similar height on the tower (and as close to each other as possible).**

This is true, but unfortunately practically difficult. The wind anemometer is located at a higher altitude to avoid tower-induced issues in determining the wind direction and thus the origin of air masses. This is especially important when studying how aerosols from local sources affect the observations. The location of the CDP near the roof decreases the wind slightly and thus the pump is used to keep the flow through the instrument as constant as possible. Because of this we are able to estimate the concentration of particles more accurately than relying on wind speed. The small-scale turbulence caused by the tower is not affecting the observed droplet size distribution, because the time the droplets spend in proximity of the tower is too short for them to adjust to the turbulence-induced changes in temperature and supersaturation.

**Page 4, line 25: "The data from these instruments is saved as one-minute averages." Change "is" to "are" i.e. "The data from these instruments are saved as one-minute averages."**

Done

**Page 5, line 1: "The CDP data were restricted to be valid only during low-level cloud events". So is this for situations where the cloud base of low level stratus clouds advecting over the region was sufficiently low to envelop the measurement tower in cloud?**

Yes, we elaborated this in the manuscript.

**Page 5, lines 1-2 ": suggest changing: ".... and the possibility for broken clouds was additionally excluded through visibility measurements." to ".... and the potential inclusion of periods of broken clouds reduced through use of visibility measurements."**

Changed.

**Page 5, line 3: insert "cloud" into: "The numerical value for the minimum cloud LWC was set to 0.02 gm-3.**

Added.

**Page 5, lines 4-5: "....  for larger LWC values the cloud droplets become so large, that the cloud can no longer be classified as non-precipitating".  This will be true unless there are cases where the droplet concentration is higher (in polluted events) and the**

**LWC is spread out over that higher droplet number.  So suggest adding a line like " ... for typical droplet number concentrations observed at the measurement site in such cases" or similar.**

Yes, we agree and changed the text accordingly. Furthermore,  for very high liquid water contents the cloud is most likely touching the surface (creating fog), or an adiabatic liquid water profile inside the cloud can no longer be assumed.

**Page 5, line 9: "...while the boundary layer depth was about 370 m from the lake level". Comment on why the BL depth was set to this value (was it validated by any measurements at the time or previously?).  Also change "....  from the lake level" to"..from above the lake level"**

This is estimated to be quite typical condition to produce cloud that does not precipitate. It is not validated as we do not have instrument available that can tell the exact location of the cloud top in all conditions. Deeper boundary layer would increase the updrafts slightly, but we do not expect this to cause any change in the analysis of results. Modelling a deeper boundary layer would also have further increased the computational cost of the LES since the computational domain has to cover the whole vertical extent of the boundary layer and also part of the free troposphere above it. One should remember that this was a relatively heavy LES run with more than 800 million grid nodes. It was computed using 512 processors in our Cray XC-30, and it took almost one hundred hours of wall-clock time.

The following text is added: "This is estimated to be quite a typical condition to produce non-precipitating cloud. We assumed that the boundary-layer depth has no remarkable influence on the simulated droplet trajectories.".

Wording is changed to: "above the lake level".

**Page 5, line 16: "using the logarithmic law of the wall for neutrally stratified conditions". Is this a correct statement (... because I am not familiar with the "wall" here) – apologies if correct! (in which case describe this a bit more)**

We admit that the original phrasing here is not the best possible choice. It is now changed to the following form which is also more descriptive: "The surface boundary condition for momentum was modelled following the Monin-Obukhov similarity theorem which assumes a constant-flux layer between the surface and the first grid layer on which the surface-tangential velocity components are solved. In this case neutrally stratified conditions were set, and the friction velocity is..." (formula for $u\_*$ follows).

**Page 6,  line 5:  delete "have" in "We have used a similar approach earlier in Romakkaniemi et al...."**

Done

**Page 8, line 11:  insert "an" in "However, as the air masses arrive at Puijo hill, they experience an updraft, which depends..."**

Done

**Page 8, line 32: suggest insert "at times" into: ... both significant and at times comparable in magnitude to the N100 trend".**

As this is the result of a statistical analysis, it is not really correct to say that the magnitude is comparable 'at times'. We therefore decided not to include this change.

**P10, line 3:  ".....typical composition measured during different campaigns (Hao et al., 2013; Portin et al., 2014; Väisänen et al., 2016) at the station."  Without reading these references in detail, was a direction dependent aerosol composition used as input here (since it was mentioned earlier that some directions included sources of aerosol.  i.e. was the aerosol composition input appropriate for the wind direction chosen that maximises the orographic effects in these sensitivity tests?**

The used composition is not exactly from the direction only as it is varying quite a lot, and we have only limited amount of aerosol composition data available compared to aerosol and cloud droplet size distribution data. However, the composition corresponds to the hygroscopicity of observed aerosol as closely as possible and also takes approximately into account the mixing state between less and more hygroscopic aerosol. The aerosol composition effect was tested with several different aerosol size distributions, and it was not found to affect the qualitative analysis.

**As an aside, since the primary control of cloud properties comes from the properties of the aerosol available as CCN, was this whole sensitivity test repeated for a quite different aerosol input? In the extreme case the shape of the aerosol distribution input to the model could change the activation pattern of CCN and result in a different cloud response in the sensitivity tests.   However I would not expect the results to change significantly. Comments?**

Yes, this was done by multiplying the aerosol concentrations with factors between 0.5 and 2, and the qualitative analysis of results did not change. Only if the size and composition between Aitken and accumulation mode was changed a lot, meaning the CCN potential of Aitken mode particles was clearly decreased, the activation of interstitial particles changed noticeably.

**Page 11, lines 10-11: " changes in the sign of the updraft are reflected as a change of sign in the supersaturation."  Maybe emphasise by saying "changes in the sign of the updraft (i.e.  changes from updraughts to downdraughts) are reflected in a change in the sign of the supersaturation (i.e. from supersaturation to subsaturation)"**

Done

**Page 12, line 8: delete superfluous "also" in ".... droplets form also inside the cloud"**

Done.

**Page 12, line 14:  delete superfluous "already" in "The in-cloud supersaturations may also be higher than at the cloud base already before the cloud parcel reaches Puijo hill"**

Done

**Anonymous Referee # 2:**

**This manuscript presents a well-performed numerical experiment to investigate topography effects on cloud droplet activation. The paper is, in general, well-written and properly structured, with no apparent scientific errors. My only major concern is related to potential implications resulting from this study. The authors only bring up the importance of considering topography effects when analyzing cloud-related measurement data at their measurement site (see abstract and conclusions). I think this work might have broader scientific implications, i.e. cloud activation studies in other hill-cloud sites or cloud droplet activation in general in environments having a complicated topography. The authors should shortly discuss this issue, for example at the end of their conclusions. My other, mainly minor concerns are detailed below.**

We will add more general discussion as suggested. In areas with complicated topography, the observed aerosol effect on cloud properties might differ from the larger area mean values and this should be acknowledged. It would be interesting also to estimate how topography is affecting the mean droplet number concentration over large area, but for this kind of study, a model with a detailed representation of both aerosol cloud interactions and surface properties would be needed.

**Page 3, lines 2-5: The statement made in these lines ( ... are mainly used to study...) needs a reference/references.**

We will add references and modify the text to be more precise.

**Page 5, lines 3-5. What was the basis of setting 0.02 g/m3 as the minimum LWC level? How was the maximum LWC level of 0.25 g/m3 found? Is it general knowledge or was it obtained from some kind of sensitivity tests specific for this orography case?**

Minimum is set because the smallest observable cloud droplet with CDP is 2 micrometers in diameter, and thus with smaller minimum LWC it becomes likely that especially in the case of high droplet concentration some droplets might stay undetected. Also the possibility of broken clouds increases when small liquid water contents are analyzed. In the larger values than 0.25 g/m3 the cloud base becomes very low, and thus it is difficult to say if it is cloud or actually fog. We clarified this in the text.

**Page 8, lines 2 and 17; caption of figure 2: "trend" is generally considered as a property of a time series and should not be used in describing other types of gradients. Please modify.**

We changed the term from 'trend' to 'effect magnitude'.

**Page 8, line 26: Please explain what is meant by standardized quantities here.**

Standardized quantities means that each of the observations are subtracted by the mean of the variable and the remainder is divided by the standard deviation. The formula is given by,

$X_{std} = (X_i - mean(x))/sd(x)$

With this, the mean of standardized variables is centered to zero and variance is 1 and thus the effects of the two predictor variables are directly comparable.

**Page 13. line 6: The reader may be a bit confused about what enhancement in CDNC means here without going back in the text. Please add a few words here to make the text more readable.**

Done.

**Figure 2:  Panels a-d differ so much from panels e-f in this figure that I would recom- mend splitting Figure 2 into two separate figures (Figs.  2 and 3).  Furthermore, the y-axises of panels e and f do not have a unit.**

This is true. We will split the figure and the text accordingly. The graphs in Figure 3 a and b (former Figure 3 e and f) depict the effect magnitude of the two predictor variables as explained in the comment above. This statistical quantity has no unit as such, but can be seen as an according (but not equal) to a derivative in the non-standardised case (as explained in the text). In the standardised case this is no longer applicable. For these reasons, a unit cannot be given.